# Attribute-Relation Guided Compositional Alignment for Weakly Supervised Referring Expression Comprehension

## Abstract

Referring expression comprehension (REC) aims to localize the object in an image described by natural language. Referring expressions often specify objects through diverse attributes and structured relations, but weakly supervised models often reduce these rich linguistic cues with coarse anchor features extracted from pre-trained detectors. The asymmetry between the expressive power of language and the limited granularity of visual features remains the core challenge for weakly supervised REC. Existing methods attempt to enrich anchors with auxiliary cues, which cannot capture diverse attributes or consistently improve instance distinctiveness. More importantly, they align text with individual anchors, which are unstructured representations unable to encode relational semantics. Capturing such structural cues requires explicitly modeling interactions among anchors. To address these limitations, we propose the Attribute–Relation guided Compositional Alignment (ARCA) framework. The proposed ARCA framework consists of two key components: (**i**) An attribute enhancer that introduces learnable attribute prototypes and, guided by subject noun chunks (*e.g.*, "a small wooden chair"), enables anchors to naturally and effectively cover diverse attribute semantics. (**ii**) A relation encoder that models inter-anchor relation representations and aligns them with full sentence embeddings, enabling the capture of structured relational cues. These two components establish a compositional alignment mechanism that enables the visual features to better match the richness and structure of language. Extensive experiments on RefCOCO, RefCOCO+, and RefCOCOg show that the proposed ARCA achieves state-of-the-art performance, demonstrating the effectiveness of compositional alignment for WREC.[1]

## 1 Introduction

Referring Expression Comprehension (REC), also known as visual grounding, aims to localize in an image the object that corresponds to a given natural language description. This task is central to many real-world applications such as human-robot interaction and visual navigation. While supervised REC methods have achieved remarkable performance, they rely on dense bounding-box annotations for training. These, however, are costly to collect at scale and limit adaptability across domains. To alleviate annotation burdens, recent works Jin et al. (2023) have shifted towards weakly-supervised REC (WREC), which leverages only paired image–text data for supervision.

Recent advances Jin et al. (2023); Luo et al. (2024); Chen et al. (2025) in WREC have focused on one-stage anchor-based frameworks, which reformulate the task as anchor-text matching optimized through contrastive learning. These methods iteratively select top-ranked anchor-text pairs as pseudo-positives and update model parameters by reinforcing these matches via contrastive loss. They effectively perform latent assignment over region proposals in an Expectation-Maximization Dempster et al. (1977) (EM)-like optimization process. This progressively refines regional vision-language alignment without box-level annotations. While existing methods frame WREC as an anchor-text matching problem, the inherent asymmetry between the visual and linguistic modalities poses a fundamental challenge. Referring expressions are often diverse, com-

---

[1]Source code and models will be released upon acceptance.

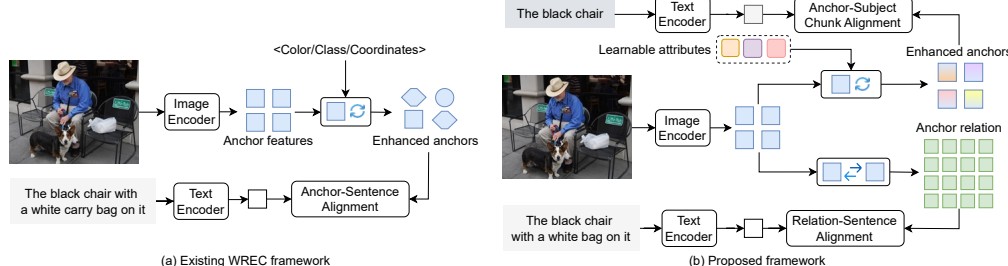

Figure 1: Comparison between existing WREC frameworks Jin et al. (2023); Luo et al. (2024); Chen et al. (2025) and the proposed framework. (a) Existing WREC frameworks rely on flat alignment, aligning text with individual anchors, APL Luo et al. (2024) enriched anchors by several manually-defined attributes. In contrast, (b) we propose a compositional alignment framework, where anchors are enhanced by learnable attributes and then aligned with subject noun chunks (*e.g.*, "The black chair"), while relation features are explicitly constructed and aligned with full sentences.

positional, and context-dependent, even for the same object instance. However, anchor features, extracted from pre-trained detection backbones, are semantically limited. These anchors generally encode category-level semantics defined by a fixed label space (*e.g.*, COCO classes Lin et al. (2014)), lacking both instance-specific discriminative power and the capacity to capture fine-grained visual attributes or contextual cues. As a result, the semantic richness of language is often inadequately grounded in the anchor space, limiting alignment accuracy under weak supervision.

Existing approaches Luo et al. (2024); Chen et al. (2025) have primarily focused on enriching individual anchor features to improve alignment. While such enrichment can benefit attribute-level grounding, most strategies rely on implicit cues (*e.g.*, features transferred from visual foundation models) Chen et al. (2025); Cheng et al. (2025) or limited manual priors (*e.g.*, spatial coordinates, category, or color) Luo et al. (2024). These signals neither comprehensively capture the diverse range of attributes nor consistently improve instance-level distinctiveness. More critically, a notable gap in existing approaches lies in their reliance on individual anchors to match textual descriptions, as illustrated in Figure 1 (a). A single anchor representation is suitable for aligning simple attribute-level descriptions, but insufficient for relational expressions that rely on interactions between objects. Relations are often directed and asymmetric. Such structural semantics cannot be captured by flat feature enrichment of individual anchors. Instead, they require explicit modeling of inter-anchor interactions to align with diverse and compositional referring expressions.

Motivated by these limitations, we propose an attribute–relation guided compositional alignment framework that explicitly enhances intra-anchor attributes and models inter-anchor relations, enabling multi-level compositional alignment beyond single anchor-text matching. More specifically, as illustrated in Figure 1 (b), at the attribute level, we propose an attribute enhancer that leverages a set of learnable attribute prototypes to enrich anchors by adaptively aggregating prototype signals according to their semantic relevance. The attribute-enhanced anchors are designed to align with the subject noun chunk through contrastive learning. This enables attribute learning to be guided by object-level linguistic units rather than coarse full-sentence semantics. Since noun chunks inherently encode diverse attributes, such as color, size, or material, they provide a natural and efficient supervisory signal that eliminates the need to manually enumerate attribute categories. A diversity constraint is further imposed on the prototypes, encouraging them to cover diverse and complementary attributes. At the relation level, we propose an anchor relation encoder to explicitly model pairwise interactions among anchors, capturing contextual dependencies and directional relations between objects. In contrast to existing approaches that align individual anchors with entire expressions, the proposed framework performs multi-level alignment: attribute-enhanced anchors are aligned with subject noun chunks, while relation features are aligned with full-sentence embeddings. This enables the model to simultaneously acquire discriminative and fine-grained intra-instance semantics and structured inter-instance relations, providing a compositional alignment mechanism that better matches the richness and structure of natural language.

The main contributions are summarized as follows: (**i**) We propose an attribute–relation guided compositional alignment (ARCA) framework for WREC, moving beyond flat anchor–text matching toward compositional alignment, addressing the gap between coarse anchor features and the

structured semantics of natural language. (**ii**) We propose an attribute enhancer that enriches anchor features via a set of learnable attribute prototypes. Guided by subject noun chunks, the enhanced anchors naturally capture fine-grained attribute semantics, thereby improving instance-level distinctiveness. (**iii**) We propose an anchor relation encoder that explicitly models pairwise anchor interactions by combining semantic and geometric cues. The resulting relation features are aligned with full-sentence embeddings, enabling the model to capture structured relational semantics. (**iv**) The proposed framework achieves the state-of-the-art performance on RefCOCO, RefCOCO+, and RefCOCOg, demonstrating the effectiveness of the proposed compositional alignment.

## 2 RELATED WORK

**Referring Expression Comprehension (REC).** REC aims to localize objects in an image based on natural language descriptions. Early supervised approaches Yang et al. (2019) relied on region-based pipelines that encode visual regions and language queries, then perform matching via joint embeddings or attention mechanisms Yu et al. (2018). More recent transformer-based models Deng et al. (2021) have further improved performance by leveraging large-scale pretraining or finetuning on large vision language models (LVLMs) Ma et al. (2024). Despite their success, these supervised methods require dense annotations at the bounding box or pixel level. This significantly limits scalability to new datasets and domains, motivating the development of weakly supervised alternatives.

**Weakly Supervised Learning (WSL).** WSL has been extensively studied to reduce annotation costs across both vision-only and vision–language tasks. In vision-only tasks, Weakly Supervised Semantic Segmentation (WSSS) Xu et al. (2022) and Weakly Supervised Object Localization (WSOL) Zhang et al. (2021) methods use only image-level labels to generate pixel-level masks or bounding boxes via class activation maps Zhou et al. (2016) and refinement strategies Ahn et al. (2019). In vision–language tasks, researchers have explored Weakly Supervised Referring Expression Comprehension (WREC) Jin et al. (2023) and Weakly Supervised Referring Expression Segmentation (WRES) Kim et al. (2023). These methods assume only paired images and referring expressions are available during training, without any box or mask annotations. Some approaches adopt pseudo-label generation Dai & Yang (2024) or adapting to Liu et al. (2023) LVLMs.

**Weakly Supervised Referring Expression Comprehension (WREC).** Early works Datta et al. (2019); Akbari et al. (2019) on WREC typically used a Multiple Instance Learning (MIL) framework Ilse et al. (2018), where all region proposals in an image were treated as a bag and instance-level representations are aggregated into bag-level representations commonly through max- or attention-pooling. A contrastive or ranking loss was then applied at the bag-text level, implicitly encouraging the model to connect the referring expression with the correct region. Some approaches Liu et al. (2019); Sun et al. (2021) introduced text reconstruction objectives, using the aggregated bag representations to reconstruct the expression. Despite effective to some extent, these methods were limited by indirect supervision to region-text correspondence or struggled by noisy negative sampling. Subsequent methods Gupta et al. (2020); Zhang et al. (2020) shifted from bag-text alignment to direct region-text contrastive learning, where highly ranked region-text pairs were leveraged in an EM-like optimization process to progressively refine alignment. This also gave rise to one-stage anchor-text frameworks Jin et al. (2023), which reformulate WREC as anchor-text matching. While one-stage frameworks improved efficiency, anchors extracted from pretrained detection models often lack instance-level distinctiveness. This has motivated research on anchor feature enrichment. APL Luo et al. (2024) explicitly incorporates auxiliary information, such as category labels and spatial coordinates, obtained from the pretrained detection model, embedding these cues into the anchor feature space. DViN Chen et al. (2025) further leverages a set of foundation models (*e.g.*, CLIP, DINO) and employs a dynamic routing mechanism to adaptively fuse multi-source visual features into each anchor, aiming to enrich the semantic diversity and contextual awareness of the anchors. A recent work, WeakMCN Cheng et al. (2025), takes a task-level perspective by jointly optimizing WREC and WRES, allowing the two tasks to provide mutual supervision and promote more consistent alignment between textual and visual modalities. In contrast to existing WREC frameworks which adopt a flat alignment strategy, where diverse expressions are all matched to individual anchors which lack the capacity to capture structured semantics, we propose a compositional alignment that separately aligns anchors with their own attribute-related text spans, and explicitly construct inter-anchor relation features to align with the full sentence, enabling structured matching that better reflects the compositional nature of language.

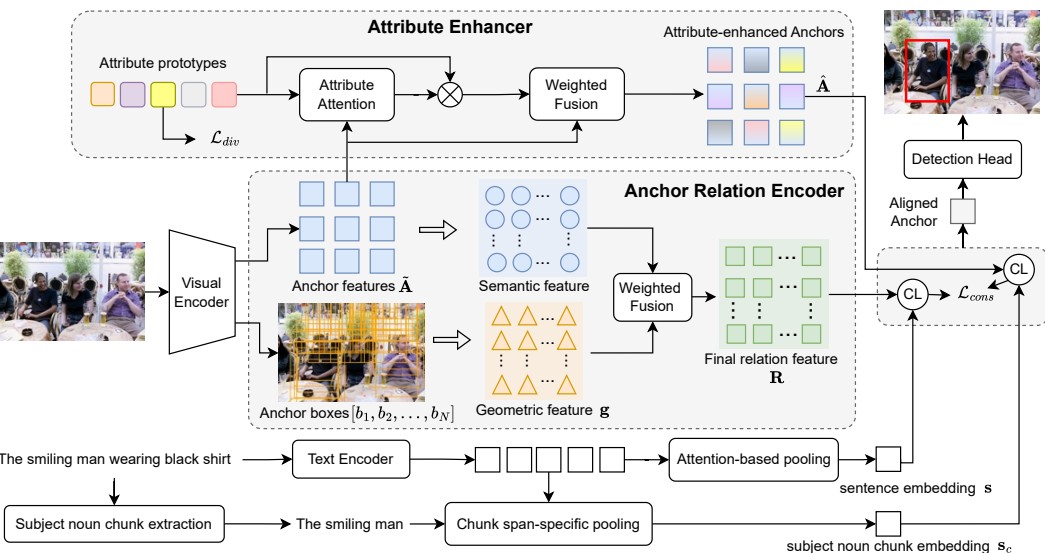

Figure 2: Overview of the proposed framework. Given an input image and referring text, a pretrained visual encoder extracts anchor features and their bounding boxes, while a text encoder produces sentence and subject noun chunk embeddings. The Attribute Enhancer enriches anchors via learnable attribute prototypes with a diversity loss, and the Anchor Relation Encoder captures pairwise semantic–geometric interactions to form relation features. A compositional alignment is performed by aligning the attribute-enhanced anchors with noun chunks and relation features with sentence embeddings through Contrastive Learning (CL). The anchor with the highest alignment score is selected and passed to a detection head for localization.

## 3 METHOD

**Overview.** As illustrated in Figure 2, the proposed framework first encodes the visual and textual modalities. The visual encoder of a pre-trained detection model is used to generate anchor-level features, while a lightweight language encoder is used to extract both subject noun chunk embeddings and full-sentence embeddings to represent local and global linguistic cues. We propose an attribute enhancer to enrich the anchor features by projecting them onto a set of learnable attribute prototypes, yielding attribute-sensitive representations that capture discriminative appearance cues. We also propose a relation encoder to model pairwise interactions among anchors, producing relation features that encode contextual dependencies between objects. Finally, we perform compositional alignment between the enriched anchor features, relation features, and language embeddings. Attribute-enhanced anchors are aligned with subject noun chunks, while relation features are aligned with full-sentence embeddings through contrastive learning. This dual alignment strategy encourages anchors to simultaneously capture local attribute semantics and global relational structures.

During inference, a referring expression is compared with anchor representations through two complementary similarity measures: subject chunks with attribute-enhanced anchors and the full sentence with relation-enriched features. The combined similarity determines the most relevant anchor, whose bounding box is then decoded by the detection head. Through this design, the proposed ARCA framework provides enriched anchor features that incorporate both attribute-level discrimination and relation-level context, thereby bridging the gap between concise and complex referring expressions across datasets.

### 3.1 FEATURE EXTRACTION BACKBONE

We use two encoders to extract visual and textual representations: a frozen YOLOv3 backbone for anchor-level visual features and a lightweight recurrent–attention encoder for textual embeddings. To further improve anchor representation, we enrich YOLOv3 features with auxiliary visual cues from visual foundation models through a dynamic routing mechanism.

**Visual Encoder.** Given an input RGB image $\mathbf{I} \in \mathbb{R}^{3 \times H \times W}$, we adopt the pre-trained DarkNet-53 backbone from YOLOv3 and take anchor-level features from the detection head, which provides multi-scale anchor representations. Following prior work Jin et al. (2023), we select the top $K$ anchors with the highest objectness scores and denote their features as $\mathbf{A} = \{\mathbf{a}_1, \mathbf{a}_2, \ldots, \mathbf{a}_K\}$, where $\mathbf{a}_i \in \mathbb{R}^d$. Motivated by prior works Chen et al. (2025); Cheng et al. (2025), we incorporate additional visual features $\mathbf{F} = \{\mathbf{f}_1, \ldots, \mathbf{f}_M\}$, where $\mathbf{f}_j \in \mathbb{R}^d$, from foundation models such as DINO v2 Oquab et al. (2024) and Depth Anything v2 Yang et al. (2024) to enhance the expressiveness of $\mathbf{A}$. Each anchor $\mathbf{a}_i$ is enriched by dynamically routing information from $\mathbf{F}$:

$$\tilde{\mathbf{a}}_i = \mathbf{a}_i + \sum_{j=1}^{M} \alpha_{ij}\, \mathbf{f}_j, \quad \alpha_{ij} = \frac{\exp\left((\mathbf{w}_j^\top \mathbf{a}_i)\right)}{\sum_{j'=1}^{M} \exp\left((\mathbf{w}_{j'}^\top \mathbf{a}_i)\right)}, \tag{1}$$

where $\alpha_{ij}$ denotes the routing weights, and $\mathbf{w}_j \in \mathbb{R}^d$ is a learnable projection vector associated with the $j$-th foundation feature. The resulting enriched anchors are denoted as $\tilde{\mathbf{A}} = \{\tilde{\mathbf{a}}_1, \ldots, \tilde{\mathbf{a}}_K\}$.

**Textual Encoder.** Given a referring expression, *e.g., "the young man standing by the table"*, we first tokenize it and map tokens into embeddings $\mathbf{E} = \{\mathbf{e}_1, \ldots, \mathbf{e}_L\}$, where $\mathbf{e}_t \in \mathbb{R}^{d_w}$. These embeddings are passed through an LSTM to obtain contextualized representations:

$$\mathbf{h}_t = \text{LSTM}(\mathbf{e}_t, \mathbf{h}_{t-1}), \quad t = 1, \ldots, L. \tag{2}$$

To capture higher-order dependencies, we apply several layers of self-attention:

$$\mathbf{z}_t = \text{SelfAttn}(\mathbf{h}_t, \mathbf{H}), \quad \mathbf{H} = \{\mathbf{h}_1, \ldots, \mathbf{h}_L\}. \tag{3}$$

From the contextualized sequence $\{\mathbf{z}_t\}_{t=1}^{L}$, we derive two levels of textual embeddings. First, a token-level attention pooling layer produces the global sentence embedding $\mathbf{s} \in \mathbb{R}^d$:

$$\beta_t = \frac{\exp(\mathbf{w}^\top \tanh(\mathbf{W}\mathbf{z}_t))}{\sum_{t'} \exp(\mathbf{w}^\top \tanh(\mathbf{W}\mathbf{z}_{t'}))}, \quad \mathbf{s} = \sum_{t=1}^{L} \beta_t \mathbf{z}_t. \tag{4}$$

Second, we extract the embedding of the subject noun chunk (*i.e., "the young man"*) by pooling the token representations: $\mathbf{s}_c = 1/(e - b + 1) \sum_{t=b}^{e} \mathbf{z}_t$, where $(b, e)$ corresponds to the chunk span.

## 3.2 Attribute–Relation Guided Compositional Alignment

To enable anchors with fine-grained attribute sensitivity and relational awareness, we introduce two complementary components: the *Attribute Enhancer*, which enriches each anchor feature with learnable attribute prototypes, and the *Anchor Relation Encoder*, which captures pairwise dependencies among anchors. The resulting features are aligned with different levels of linguistic representations through contrastive objectives.

**Attribute Enhancer.** Given the set of enriched anchor features $\tilde{\mathbf{A}} = \{\tilde{\mathbf{a}}_1, \ldots, \tilde{\mathbf{a}}_K\}$ obtained from the backbone, we introduce $N$ learnable attribute prototypes $\mathbf{P} = \{\mathbf{p}_1, \ldots, \mathbf{p}_N\}$ to further enhance the awareness of the attributes that are shared by the entire dataset, where $\mathbf{p}_j \in \mathbb{R}^d$. For each anchor $\tilde{\mathbf{a}}_i$, we compute its similarity to the prototypes as weights $\gamma_{ij}$ and derive its specific attribute features $\mathbf{a}_i^{\text{attr}}$ via as a weighted combination of prototypes:

$$\mathbf{a}_i^{\text{attr}} = \sum_{j=1}^{N} \gamma_{ij}\, \mathbf{p}_j, \quad \gamma_{ij} = \frac{\exp(\mathbf{p}_j^\top \tilde{\mathbf{a}}_i)}{\sum_{j'=1}^{N} \exp(\mathbf{p}_{j'}^\top \tilde{\mathbf{a}}_i)}. \tag{5}$$

The attribute-enhanced anchors $\hat{\mathbf{A}} = \{\hat{\mathbf{a}}_1, \ldots, \hat{\mathbf{a}}_K\}$ are obtained by adaptively combining the original anchors and their attribute features via a gated fusion:

$$\hat{\mathbf{a}}_i = \mathbf{g}_i \odot \mathbf{a}_i^{\text{attr}} + (1 - \mathbf{g}_i) \odot \tilde{\mathbf{a}}_i, \quad \mathbf{g}_i = \sigma\big(\text{MLP}([\tilde{\mathbf{a}}_i \,\|\, \mathbf{a}_i^{\text{attr}}])\big), \tag{6}$$

where $\sigma(\cdot)$ denotes the sigmoid activation and $[\cdot \| \cdot]$ represents concatenation. The attribute-enhanced anchors are finally processed by a layer normalization. To encourage the prototypes to capture diverse attribute directions, we regularize them with an orthogonality constraint:

$$\mathcal{L}_{\text{div}} = \left\| \mathbf{B}\mathbf{B}^\top - \mathbf{I} \right\|_F^2, \tag{7}$$

where $\mathbf{B} = \mathrm{normalize}(\mathbf{P}, \ell_2)$, $\mathbf{I}$ denotes the identity matrix, and $\|\cdot\|_F$ denotes the Frobenius norm.

**Anchor Relation Encoder.** While attribute enhancement improves intra-anchor discrimination, it does not capture relations among objects. We thus construct pairwise relation features $\mathbf{r}_{ij}$ between anchors based on their semantic and geometric features. For any anchor pair $(i, j)$, we define

$$\mathbf{r}_{ij} = w_{ij} \cdot \mathrm{MLP}\big([\tilde{\mathbf{a}}_i \,\|\, \tilde{\mathbf{a}}_j \,\|\, \delta_{ij}]\big), \quad \delta_{ij} = \mathrm{MLP}(\mathbf{g}_{ij}), \tag{8}$$

where $\mathbf{g}_{ij}$ denotes an 8-dimensional geometric descriptor encoding relative position, scale, overlap, and orientation: $[\Delta x, \Delta y, \log(w_j/w_i), \log(h_j/h_i), \mathrm{IoU}, d, \cos\theta, \sin\theta]$. The learnable weight $w_{ij} = \sigma\big(\mathrm{MLP}([\tilde{\mathbf{a}}_i \,\|\, \tilde{\mathbf{a}}_j \,\|\, \delta_{ij}])\big)$ adaptively modulates the strength of each relation feature, suppressing noisy while preserving meaningful interactions.

**Compositional Alignment.** We align the two enhanced feature sets with different levels of textual representations. More specifically, attribute-enhanced anchors are aligned with the subject noun chunk embedding $\mathbf{s}_c$, while the relation features are aligned with the global sentence embedding $\mathbf{s}$. The alignment is optimized with InfoNCE contrastive objectives:

$$\mathcal{L}_{\mathrm{attr}} = -\log \frac{\exp(\hat{\mathbf{a}}_{i*}^\top \mathbf{s}_c / \tau)}{\sum_{i=1}^{K} \exp(\hat{\mathbf{a}}_i^\top \mathbf{s}_c / \tau)}, \tag{9}$$

$$\mathcal{L}_{\mathrm{rel}} = -\log \frac{\exp(\bar{\mathbf{r}}_{i*}^\top \mathbf{s} / \tau)}{\sum_{i=1}^{K^2} \exp(\bar{\mathbf{r}}_i^\top \mathbf{s} / \tau)}, \tag{10}$$

where $i^*$ denotes the pseudo ground-truth anchor index during training, and $\tau$ is a temperature parameter. We also enforce a hierarchical consistency between subject-level and sentence-level matching. To encourage the anchors that are highly ranked by subject-level alignment to be also assigned high probability by relation-level alignment, we penalize when relation scores fall outside the subject top-$k$ set. The consistency loss is then defined as:

$$\mathcal{L}_{\mathrm{cons}} = -\log m, \quad m = \sum_{i \in \mathrm{TopK}(\mathbf{s}_A)} \mathbf{s}_R(i), \tag{11}$$

where $\mathbf{s}_A \in \mathbb{R}^K$ denote the similarity scores between anchors and the subject embedding $\mathbf{s}_c$, and $\mathbf{s}_R \in \mathbb{R}^K$ denote the similarity scores between relation-enriched anchors and the full sentence embedding $\mathbf{s}$. The overall training objective loss function is:

$$\mathcal{L} = \mathcal{L}_{\mathrm{attr}} + \mathcal{L}_{\mathrm{rel}} + \mathcal{L}_{\mathrm{div}} + \mathcal{L}_{\mathrm{cons}}. \tag{12}$$

### 3.3 GROUNDING INFERENCE

Given a referring expression, we predict the target object by aligning it with both relation-aware and attribute-enhanced visual anchors. The inference process consists of three steps: a relation-based anchor filtering stage, an attribute-based re-ranking, and final prediction.

**(1) Relation-based anchor filtering.** We first match the full-sentence embedding $\mathbf{s}$ with the pairwise relation features $\{\mathbf{r}_{ij}\}$. A symmetric similarity score for each anchor is computed as:

$$\mathrm{Sim}_i^{\mathrm{rel}} = \frac{1}{2}\left(\log \sum_j \exp(\mathbf{r}_{ij}^\top \mathbf{s}) + \log \sum_j \exp(\mathbf{r}_{ji}^\top \mathbf{s})\right). \tag{13}$$

This yields a confidence distribution over anchors based on their contextual interactions. We select the top-$k$ anchors with the highest relation-based scores for further refinement.

**(2) Attribute-based re-ranking.** For each of the top-$k$ anchors, we compute its similarity to the subject noun chunk embedding $\mathbf{s}_c$ using the attribute-enhanced anchor features $\{\hat{\mathbf{a}}_i\}$:

$$\mathrm{Sim}_i^{\mathrm{attr}} = \hat{\mathbf{a}}_i^\top \mathbf{s}_c. \tag{14}$$

We then fuse the two similarity measures:

$$\mathrm{Score}_i = \alpha \cdot \mathrm{Sim}_i^{\mathrm{rel}} + \beta \cdot \mathrm{Sim}_i^{\mathrm{attr}}, \tag{15}$$

where $\alpha = 0.6$, $\beta = 0.4$ balance the contributions of contextual and attribute cues.

**(3) Final prediction.** The anchor with the highest fused score is selected as the target. Its bounding box is decoded via the detection head to produce the final grounding output.

| Task | Methods | Sup. | RefCOCO | | | RefCOCO+ | | | RefCOCOg |
| --- | --- | --- | --- | --- | --- | --- | --- | --- | --- |
| | | | val | testA | testB | val | testA | testB | val |
| REC | SimVG-B Dai et al. (2024) | B | 87.63 | 90.22 | 84.04 | 78.65 | 83.36 | 71.82 | 78.81 |
| | OneRef-B Xiao et al. (2024) | B | 88.75 | 90.95 | 85.34 | 80.43 | 86.46 | 74.26 | 83.68 |
| | RefFormer-B Wang et al. (2024) | B | 86.52 | 90.24 | 81.42 | 76.58 | 83.69 | 67.38 | 77.80 |
| WREC | ARN Liu et al. (2019) | T | 32.17 | 35.25 | 30.28 | 32.78 | 34.35 | 32.13 | 33.09 |
| | IGN Zhang et al. (2020) | T | 34.78 | 37.64 | 32.59 | 34.29 | 36.91 | 33.56 | 34.92 |
| | DTWREG Sun et al. (2021) | T | 38.35 | 39.51 | 37.01 | 38.19 | 39.91 | 37.09 | 42.54 |
| | RefCLIP Jin et al. (2023) | T | 60.36 | 58.58 | 57.13 | 40.39 | 40.45 | 38.86 | 47.87 |
| | APL Luo et al. (2024) | T | 64.51 | 61.91 | _63.57_ | 42.70 | 42.84 | 39.80 | 50.22 |
| | WeakMCN Cheng et al. (2025) | T | _69.20_ | 69.88 | 62.63 | 51.90 | 57.33 | 43.10 | 54.62 |
| | DViN Chen et al. (2025) | T | 67.67 | _70.90_ | 59.39 | _52.54_ | _57.52_ | **45.31** | _55.04_ |
| | ARCA (ours) | T | **71.06** | **72.60** | **65.55** | **54.17** | **60.22** | _44.63_ | **56.99** |

Table 1: Performance comparison with the state-of-the-art methods in REC and WREC on Ref-COCO, RefCOCO+, and RefCOCOg. B and T denote bounding box and text supervision (Sup.), respectively. The best and the second best results are **bold** and underlined.

## 4 EXPERIMENTS

### 4.1 EXPERIMENTAL SETTINGS

**Datasets.** Following the common practice Jin et al. (2023); Luo et al. (2024); Chen et al. (2025), we evaluate our method on three widely used referring expression comprehension benchmarks: RefCOCO Yu et al. (2016), RefCOCO+ Yu et al. (2016), and RefCOCOg Nagaraja et al. (2016), all derived from the MSCOCO dataset Lin et al. (2014) with object-level referring expressions. We use the standard splits for fair comparison. More dataset details are provided in the Appendix.

**Evaluation metric.** IoU@0.5 is used to measure the accuracy of the predicted bounding boxes whose Intersection over Union (IoU) with the ground-truth bounding boxes are above 0.5.

**Implementation details.** Following prior work Jin et al. (2023); Chen et al. (2025); Cheng et al. (2025), we adopt YOLOv3 as the detection backbone, and use DINOv2 Oquab et al. (2024) and Depth Anything v2 Yang et al. (2024) as additional visual features. Text embeddings are initialized with GloVe Pennington et al. (2014). More details are provided in the Appendix.

### 4.2 COMPARISON WITH STATE-OF-THE-ART

Table 1 reports the state-of-the-art results under both fully supervised REC and weakly supervised REC (WREC). While fully supervised methods achieve the highest scores, thanks to box-level annotations, the proposed ARCA, trained without such supervision, shows competitive results (*e.g.*, 71.06% and 72.60% on RefCOCO *val* and *testA*), narrowing the gap between supervised and weakly supervised paradigms. Compared to DViN, the strongest WREC method, the proposed ARCA consistently outperforms or achieves comparable results across all splits of three datasets, particularly showing significant gains of +3.39% on RefCOCO *val*, +6.61% on RefCOCO *testB*, +2.70% on RefCOCO+ *testA*, and +1.95% on RefCOCOg *val*. The proposed ARCA surpasses WeakMCN, a strong multi-task learning method, by notable margins of +2.72% and +2.93% on RefCOCO *testA* and *testB*, +2.27% and +2.89% on RefCOCO+ *val* and *testA*, and +2.37% on RefCOCOg *val*. These robust gains confirm the effectiveness and generalization ability of the proposed ARCA for WREC.

### 4.3 ABLATION STUDY

**Effect of Attribute Enhancer and Relation Encoder.** As shown in Table 2, adding either the attribute enhancer or the relation encoder consistently improves performance over the baseline. On RefCOCO, where expressions are often short, the attribute enhancer brings strong improvements (*e.g.*, +3.34% on *val* and +4.36% on *testB*). In contrast, on RefCOCOg, which contains long and unconstrained descriptions, relation modeling is more beneficial (+3.23% gain on *val*), reflecting the importance of capturing inter-object relations in free-form language. On RefCOCO+, where referring expressions are longer and less position-driven, single modules bring only modest gains, but combining them yields larger improvements. Similar trends are observed across RefCOCO and RefCOCOg, with the largest boost (+5.91%) is achieved when both modules are combined on the

| Methods | Align. | RefCOCO | | | RefCOCO+ | | | RefCOCOg |
|---|---|---|---|---|---|---|---|---|
| | | val | testA | testB | val | testA | testB | val |
| Baseline (DINO + Depth) | FA | 66.59 | 69.10 | 59.94 | 51.22 | 58.28 | 41.62 | 51.08 |
| + AttrEnhancer | FA | 69.93 | 71.52 | 64.30 | 53.50 | 58.52 | 43.08 | 53.31 |
| + AnchorRel | CA | 69.22 | 71.98 | 63.47 | 52.45 | 58.89 | 41.97 | 54.31 |
| + AttrEnhancer + AnchorRel | CA | **71.06** | **72.60** | **65.55** | **54.17** | **60.22** | **44.63** | **56.99** |
| Δ | - | ↑ 4.47 | ↑ 3.50 | ↑ 5.61 | ↑ 2.95 | ↑ 1.94 | ↑ 3.01 | ↑ 5.91 |
| Baseline (DINO + CLIP) | FA | 63.71 | 65.25 | 56.09 | 48.98 | 54.16 | 40.15 | 50.53 |
| + AttrEnhancer + AnchorRel | CA | **68.90** | **70.14** | **60.43** | **52.77** | **58.71** | **42.46** | **53.82** |
| Δ | - | ↑ 5.19 | ↑ 4.89 | ↑ 4.34 | ↑ 3.79 | ↑ 4.55 | ↑ 2.31 | ↑ 3.29 |
| WeakMCN* Cheng et al. (2025) | FA | 68.55 | 70.78 | 62.00 | 51.48 | 56.92 | 41.75 | 53.44 |
| + AttrEnhancer + AnchorRel | CA | **71.12** | **72.16** | **66.10** | **53.79** | **59.12** | **43.94** | **55.92** |
| Δ | - | ↑ 2.57 | ↑ 1.38 | ↑ 4.10 | ↑ 2.31 | ↑ 2.20 | ↑ 2.19 | ↑ 2.48 |
| WeakMCN† Cheng et al. (2025) | FA | 69.20 | 69.88 | 62.63 | 51.90 | 57.33 | 43.10 | 54.62 |
| + AttrEnhancer + AnchorRel | CA | **70.54** | **71.43** | **64.89** | **53.23** | **59.57** | **44.04** | **55.94** |
| Δ | - | ↑ 1.34 | ↑ 1.55 | ↑ 2.26 | ↑ 1.33 | ↑ 2.24 | ↑ 0.96 | ↑ 1.32 |

Table 2: Effect of the proposed ARCA framework and its main components. Align.: alignment type. FA: flat alignment. CA: compositional alignment. Δ: performance gain of the proposed method compared to the baseline and WeakMCN Cheng et al. (2025), following which we report results using ViT-tiny- and ViT-small- based SAM models denoted by ∗ and †, respectively.

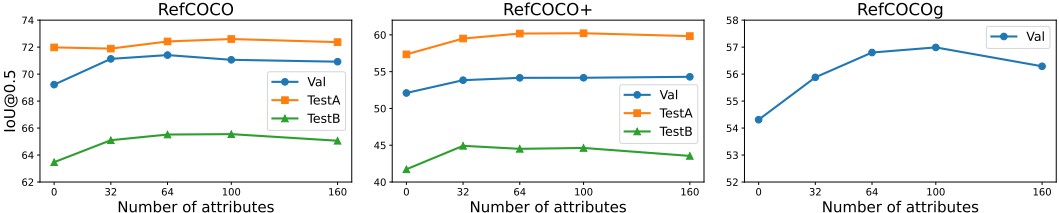

Figure 3: Effect of varying the number of attribute prototypes.

RefCOCOg *val* set. These results suggest that attributes and relations play complementary roles, and their synergy is particularly important for handling longer, composition-heavy expressions.

**Compositional Alignment *v.s.* Flat Alignment.** Table 2 also shows that when the proposed two modules are combined under Compositional Alignment (CA), the gains are further improved. CA achieves the best results across all datasets, outperforming Flat Alignment (FA) by a substantial margin (up to +5.91% on the *val* set of RefCOCOg). Importantly, this trend holds even when integrating our modules into WeakMCN Cheng et al. (2025), a recent multi-task method: CA consistently yields higher accuracy than FA. These results demonstrate that the proposed method provides a general and complementary improvement, enabling effective grounding performances.

**Baseline Analysis.** Motivated by DViN Chen et al. (2025) and WeakMCN Cheng et al. (2025), we enrich the original anchor features with additional representations extracted from visual foundation models via a dynamic routing mechanism. By default, we use DINOv2 features, which have been widely demonstrated to be semantically rich. We also introduce DepthAnything features to provide depth cues. Table 2 shows that our proposed modules consistently bring significant gains across different baselines, *e.g.*, +5.61% and +4.34% with "DINO+Depth" and "DINO+CLIP", respectively, on RefCOCO *testB*. Comparing two baselines, "DINO+Depth" is consistently stronger than "DINO+CLIP". This suggests that geometric depth cues provide more complementary information to DINO's semantic features than CLIP's visual embeddings, which are semantically aligned but less effective in capturing fine-grained structural distinctions under weak supervision.

**Effect of the Number of Attribute Prototypes.** Figure 3 shows the effect of varying the number of attribute prototypes in the attribute enhancer. The performance generally improves as the number of attribute prototypes increases across all datasets up to 100, suggesting that richer prototype sets capture more diverse and fine-grained attributes. The gains are most noticeable on RefCOCOg with the longer and more descriptive expressions, where capturing subtle attributes is crucial. However, the performance declines on most datasets when learning beyond 100 prototypes. This is likely due to the redundancy and overfitting. Overall, a moderate prototype size (*e.g.*, 100, as used in our main experiments) provides the best balance between diversity and distinctiveness.

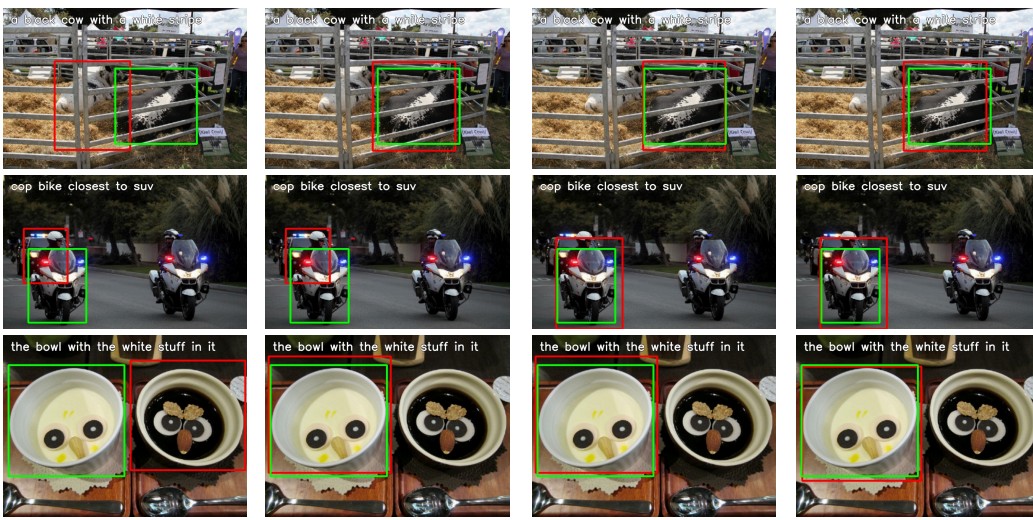

Figure 4: Qualitative results. From left to right, each column corresponds to the baseline, baseline with AttrEnhancer, baseline with AnchorRel, and the full ARCA model, respectively. Red boxes: predictions. Green boxes: GT. More visualizations are provided in the Appendix.

**Effect of different regularization losses.** As shown in Table 3, applying diversity loss ($\mathcal{L}_{\text{div}}$) alone improves performance from 64.59% to 65.48% on RefCOCO *testB*, which contains more non-human related expressions compared to other splits that are dominated by human-centric descriptions. This indicates that promoting diverse attribute prototypes is especially beneficial for

Table 3: Effect of different regularization losses.

| $\mathcal{L}_{\text{div}}$ | $\mathcal{L}_{\text{cons}}$ | RefCOCO | | RefCOCO+ | |
| --- | --- | --- | --- | --- | --- |
| | | testA | testB | testA | testB |
| | | 71.88 | 64.59 | 58.47 | 43.58 |
| ✓ | | 71.75 | 65.48 | 59.27 | 43.51 |
| | ✓ | 72.21 | 64.46 | 59.48 | 44.39 |
| ✓ | ✓ | **72.60** | **65.55** | **60.22** | **44.63** |

handling broader and less familiar categories. In contrast, the consistency loss ($\mathcal{L}_{\text{cons}}$), which enforces alignment consistency between anchor–noun and relation–sentence matching, is particularly effective on RefCOCO+ (*i.e.,*, from 48.47% to 59.48% on *testA* and from 43.58% to 44.39% on *testB*), where expressions are longer and rich in relational semantics. Combining these two regularizations achieves the best results across all datasets. This demonstrates their effectiveness in strengthening the generalization ability of the framework across diverse linguistic scenarios.

**Qualitative Results.** Figure 4 shows how the proposed modules contribute to grounding performance under different types of referring expressions. For expressions that primarily describe the object itself, the attribute enhancer significantly improves localization accuracy by enlarging the anchors' instance-level distinctiveness with learned attribute cues, *e.g.*, in the first example "a black cow with white stripe", it helps the model focus on the cow with the distinctive stripe. In contrast, when expressions emphasize relations between objects (*e.g.*, the second row, "cop bike closest suv"), the Anchor Relation module proves highly effective, correctly disambiguating the target "bike" among multiple visually similar candidates by explicitly modeling inter-anchor dependencies. The full ARCA model, which integrates both modules, achieves the most precise localization by capturing both attribute-level cues and inter-object relations.

## 5 CONCLUSION

We proposed ARCA, an attribute–relation guided compositional alignment framework, for weakly supervised referring expression comprehension. Unlike prior approaches that focus on intra-anchor enrichment and align them directly with full expressions, the proposed ARCA explicitly disentangles semantics into attributes and relations, aligning attribute-enhanced anchors with subject noun chunks and explicitly constructed anchor relation features with full sentences, respectively. Through this design, ARCA effectively bridges the structural gap between visual anchors and the diverse and relational semantics of natural language. Extensive experiments on RefCOCO, RefCOCO+, and RefCOCOg demonstrate that the proposed ARCA consistently outperforms existing WREC methods, highlighting the effectiveness of compositional alignment in the weakly supervised setting.

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

# A APPENDIX

## A.1 ADDITIONAL EXPERIMENTAL DETAILS

**Dataset Details.** RefCOCO contains 142,210 expressions referring to 50,000 objects in 19,994 images. It contains relatively short and direct expressions, which include more spatial descriptions. RefCOCO+ includes 141,564 expressions referring to 49,856 objects in 19,992 images, but explicitly discourages the use of spatial terms during annotation, thus having more appearance-based descriptions. RefCOCOg incudes 104,560 expressions, which are longer and more complex than those in RefCOCO and RefCOCO+. It covers 54,822 objects in 26,711 images, with more compositional descriptions about both appearance and spatial information.

**Implementation Details.** Following prior work Jin et al. (2023); Chen et al. (2025); Cheng et al. (2025), we used the pretrained YOLOv3 as the detection network. Following DViN Chen et al. (2025) and WeakMCN Cheng et al. (2025), we incorporate pretrained visual foundation models to enrich anchor features. More specifically, we use DINOv2-base Oquab et al. (2024) model, and for baseline comparisons, we the CLIP ViT-base Radford et al. (2021) vision encoder. We additionally use the Depth Anythingv2-small Yang et al. (2024) model. We initialize the text embeddings with GloVe Pennington et al. (2014) pretrained weights and set the maximum sequence length to 15. We use the SpaCy python library to extract the subject noun chunk in each referring expression. The training image size is set to $416 \times 416$. The training starts with a 3-epoch warm-up step, with the learning rate linearly increasing from 1e-4 to 1e-3. After the warm-up, the learning rate starts at 1e-3 with a cosine decay schedule. The model is trained for a total of 25 epochs. All experiments are conducted on a single NVIDIA RTX 3090 GPU.

## A.2 ADDITIONAL QUALITATIVE RESULTS.

Figure 5 provides additional qualitative examples to illustrate the effect of the proposed modules. For attribute-focused descriptions (*e.g.*, "white donut", "cup filled with beverage", "brownie with white top"), the Attribute Enhancer consistently improves localization by enabling anchors to be more discriminative with respect to fine-grained attributes, whereas the baseline often confuses the target with visually similar instances. For relation-based expressions (*e.g.*, "the man on the skateboard" and the action-oriented "kid stealing treats"), the Anchor Relation module correctly localizes the target instance by modeling inter-object dependencies, while the baseline or adding Attribute Enhancer tends to mis-cover nearby or co-occurring objects. In a more complex case of "green plant behind a table visible behind a lady", the baseline mislocalizes the table, the Attribute Enhancer identifies a plant but not the correct instance, while the Anchor Relation module correctly localizes the target plant, as does the full model. These results highlight that attributes and relations play complementary roles, and their integration enables robust performance across diverse expression types.

## A.3 USE OF LARGE LANGUAGE MODELS (LLMS)

We used a large language model (ChatGPT) solely as a writing assist tool to polish the grammar, clarity, and readability of the manuscript. The conceptualization of the research, technical development, experimental design, implementation, and analysis were carried out entirely by the authors. The authors take full responsibility for all content in this paper.

(1) Referring expression: *white donut*

(2) Referring expression: *cup filled with beverage*

(3) Referring expression: *brownie with white top*

(4) Referring expression: *kid stealing treats*

(5) Referring expression: *the man on the skateboard*

(6) Referring expression: *green plant behind a table visible behind a lady*

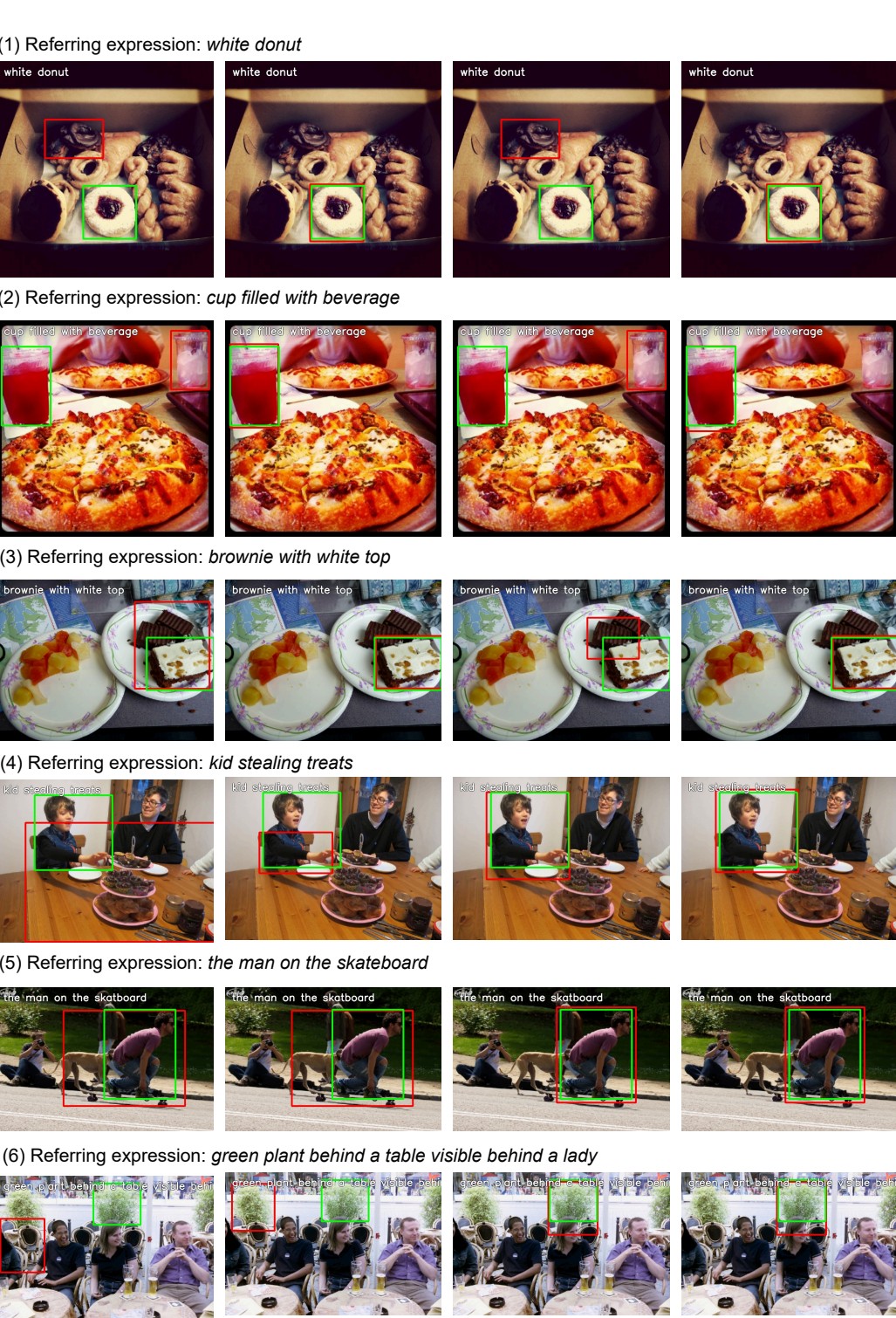

Figure 5: Qualitative results. From left to right, each column corresponds to the baseline, baseline with AttrEnhancer, baseline with AnchorRel, and the full ARCA model, respectively. Red boxes: predictions. Green boxes: GT.

