# OpenReview forum: "Attribute-Relation Guided Compositional Alignment for Weakly Supervised Referring Expression Comprehension"
_ICLR.cc/2026/Conference — ICLR 2026 Conference Withdrawn Submission_

### Official Review · Reviewer_6bne · 2025-10-19

**Soundness:** 4
**Presentation:** 4
**Contribution:** 3
**Rating:** 8
**Confidence:** 4

**Summary:**

This paper replaces flat text, anchor matching with compositional alignment: (i) learn attribute prototypes to enhance anchors and align them with subject noun chunks; (ii) explicitly encode pairwise anchor relations and align them with the full sentence, yielding consistent SOTA on RefCOCO/+/g under weak supervision. for the results,  Table 1 shows ARCA outperforms WREC methods on most rec dataset splits (e.g., RefCOCO val/testA/testB: 71.06/72.60/65.55; RefCOCOg val: 56.99).

**Strengths:**

1. This paper addresses a real gap with a clean decomposition in the weakly rec field, abd splitting alignment into (subject‑chunk with attributes) and (sentence with relations) is an intuitive fix for flat alignment’s limits; the pipeline in Fig. 2 is clearly designed.

2. Consistent empirical gains across three datasets, multiple splits, even when integrated into WeakMCN the compositional alignment still improves accuracy.

3. The proposed method ARCA demonstrates a sota performance compared with previous works and methods.

4. The paper is well written and easy to read.

5. In table3, the regularization choices have interpretable effects. Where I can see Ldiv helps non‑human/testB cases (diverse categories); Lcons helps relation‑rich RefCOCO+, further prove the effectiveness of the method.

**Weaknesses:**

1. Parser brittleness is under‑analyzed. The design hinges on subject noun‑chunk extraction (SpaCy). No robustness study for parsing errors, multi‑head subjects, or expressions with implicit subjects (common in RefCOCO+).

2. Complexity of relation modeling. Pairwise relations are O(K²) over anchors; practical K and runtime/memory are not reported. A compute/latency table would clarify deployability.

**Questions:**

1. Can you visualize top images/anchors per prototype and map them to common attribute words? Even a coarse audit (color/size/material/texture) would make the “attribute” claim concrete.

2. What are K and pair counts per image, and the wall‑clock for relation encoding? plz do a efficient analysis.

---

### Official Review · Reviewer_wSQi · 2025-10-28

**Soundness:** 2
**Presentation:** 3
**Contribution:** 2
**Rating:** 4
**Confidence:** 4

**Summary:**

This paper proposes a framework for Weakly Supervised Referring Expression Comprehension (WREC). Basically, a compositional alignment architecture is designed with both an anchor attribute enhancer and an anchor relation encoder to make better alignment between anchor and text under weakly supervised settings. In inference stage, given a referring expression, the algorithm performs relation-based anchor filtering and attribute-based re-ranking sequentially to get a final bounding box prediction. Experiment shows that the proposed algorithm achieves state-of-the-art performance on a set of public benchmarks including RefCOCO, RefCOCO+, and RefCOCOg.

**Strengths:**

1. The idea of adding inter-anchor relationship for better anchor-text alignment is reasonable
2. The figure 4 and figure 5 clearly demonstrate how the proposed modules contribute to improved grounding performance under different types of referring expressions.
3. The ablation study in table 2 clearly shows the performance gain of the proposed module of attribute enhancer and relation encoder.

**Weaknesses:**

1. Although the proposed algorithm includes sophisticated implementations such as multiple learnable attribute prototypes, multi-stage inference time filtering & re-ranking, the resulted overall performance improvement is not very impressive, i.e in Table 1. +1.95% over DViN on RefCOCOg-val and -0.68% over DViN on RefCOCO+ testB dataset.
2. A more in-depth discussion or experiment of computational overhead compared to flat alignment strategies would be beneficial.
3. Since using VLM for REC is popular nowdays, it would be beneficial to add recent advancement in VLM and their comparisons with proposed algorithm in the section of Related Works

**Questions:**

1. Part of the current ablation study in table 2 is based on WeakMCN, how about the ablation study based on DViN (i.e performance of DViN, and DViN+ AttrEnhancer + AnchorRel)?
2. Does the proposed AttrEnhancer + AnchorRel generalizable to other REC benchmarks, such as ReferItGame and Flickr30k Entities?

---

### Official Review · Reviewer_2B8Z · 2025-10-31

**Soundness:** 3
**Presentation:** 3
**Contribution:** 2
**Rating:** 2
**Confidence:** 4

**Summary:**

In this paper, the authors propose an Attribute–Relation guided Compositional Alignment (ARCA) framework for Weakly Supervised Referring Expression Comprehension (REC). ARCA adds attribute information to enhance anchors and model the relationships among anchors to improve the performance. The results achieved significant improvements.

**Strengths:**

1. The accuracies on many datasets are significantly improved.
2. Many ablation studies show the effectiveness of each module.

**Weaknesses:**

1. Adding attributes and relations is widely used in referring expression comprehension (REC) and related tasks, such as (but not limited to) [A-J].

2. How to extract the subject noun chunk? If the authors use language parsing tools, it is the same as the previous attribute/relation-based REC works.

3. The relations between anchors are generated by semantics and locations, which are very similar to scene graph generation. This approach is often commonly used in REC.

4. The authors calculate the relations between every anchor pair. Isn't it too expensive? It would be great to report the computational costs, such as FLOPs or training/testing time of the proposed method, and compare them with other methods.

5. As many efficient scene graph generation methods are proposed, it would be better to try these methods rather than calculating the relations between every anchor pair.


[A] Liu J, Wang L, Yang M H. Referring expression generation and comprehension via attributes[C]//Proceedings of the IEEE International Conference on Computer Vision. 2017: 4856-4864.

[B] Yu L, Lin Z, Shen X, et al. Mattnet: Modular attention network for referring expression comprehension[C]//Proceedings of the IEEE conference on computer vision and pattern recognition. 2018: 1307-1315.

[C] Zhang C, Li W, Ouyang W, et al. Referring expression comprehension with semantic visual relationship and word mapping[C]//Proceedings of the 27th ACM International Conference on Multimedia. 2019: 1258-1266.

[D] Wang Y, Ji Z, Wang D, et al. Towards unsupervised referring expression comprehension with visual semantic parsing[J]. Knowledge-Based Systems, 2024, 285: 111318.

[E] Lyu F, Feng W, Wang S. vtGraphNet: Learning weakly-supervised scene graph for complex visual grounding[J]. Neurocomputing, 2020, 413: 51-60.

[F] Liu Y, Wan B, Ma L, et al. Relation-aware instance refinement for weakly supervised visual grounding[C]//Proceedings of the IEEE/CVF conference on computer vision and pattern recognition. 2021: 5612-5621.

[G] Xiao F, Sigal L, Jae Lee Y. Weakly-supervised visual grounding of phrases with linguistic structures[C]//Proceedings of the IEEE Conference on Computer Vision and Pattern Recognition. 2017: 5945-5954.

[H] Lin P, Li R, Ji Y, et al. Triple alignment strategies for zero-shot phrase grounding under weak supervision[C]//Proceedings of the 32nd ACM International Conference on Multimedia. 2024: 4312-4321.

[I] Zhang C, Li W, Ouyang W, et al. Referring expression comprehension with semantic visual relationship and word mapping[C]//Proceedings of the 27th ACM International Conference on Multimedia. 2019: 1258-1266.

[J] Yang Z, Liu Y, Lin J, et al. Boosting weakly supervised referring image segmentation via progressive comprehension[J]. Advances in Neural Information Processing Systems, 2024, 37: 93213-93239.

**Questions:**

Please see the weaknesses.

---

### Official Review · Reviewer_Vtwz · 2025-10-31

**Soundness:** 2
**Presentation:** 2
**Contribution:** 2
**Rating:** 2
**Confidence:** 1

**Summary:**

This paper proposes the Attribute–Relation guided Compositional Alignment (ARCA) framework to address the mismatch between rich linguistic semantics and coarse visual features in weakly supervised referring expression comprehension. By introducing learnable attribute prototypes and a relation encoder for modeling inter-anchor relations, ARCA achieves compositional alignment between vision and language, leading to state-of-the-art performance on RefCOCO, RefCOCO+, and RefCOCOg datasets.

**Strengths:**

1. This paper is easy to follow
2. This experimental result shows that this method is competitive.

**Weaknesses:**

1. The novelty of this paper is limited. There already exists a weakly supervised REC method [1] that considers attribute–relation information among objects. Please clarify the primary differences between your approach and [1].
2. Regarding text embeddings, several existing methods extract subjects and attributed objects from expressions for matching, such as [2] and [3]. How does your approach differ from or improve upon these methods?
3. How are the geometric features of the object candidates’ bounding boxes represented, and what is the basis for this representation?
4. Please provide examples or analysis of the failure cases of your method.
[1] Yongfei Liu, Bo Wan, Lin Ma, and Xuming He. Relation-aware instance refinement for weakly supervised visual grounding. In CVPR, 2021
[2] J Ke, J Wang, JC Chen, IH Jhuo, CW Lin, YY Lin, CLIPREC: Graph-based domain adaptive network for zero-shot referring expression comprehension, IEEE Transactions on Multimedia 2023, 26, 2480-2492
[3]Sibei Yang, Guanbin Li, Yizhou Yu, Graph-Structured Referring Expressions Reasoning in The Wild, CVPR, 2020

**Questions:**

see above-mentioned details

---

### Note · Authors · 2025-11-14

I have read and agree with the venue's withdrawal policy on behalf of myself and my co-authors.